# Ambient Environmental Ozone and Variation of Fractional Exhaled Nitric Oxide (FeNO) in Hairdressers and Healthcare Workers

**DOI:** 10.3390/ijerph20054271

**Published:** 2023-02-28

**Authors:** Tonje Trulssen Hildre, Hilde Heiro, Ingvill Sandven, Bato Hammarström

**Affiliations:** Environmental and Occupational Medicine, Department of Pulmonary Medicine, Division of Medicine, Oslo University Hospital, 0424 Oslo, Norway

**Keywords:** environmental exposures, asthma, FeNO, hairdressers, healthcare workers, air quality, ozone

## Abstract

Fractional exhaled nitric oxide (FeNO) is a breath-related biomarker of eosinophilic asthma. The aim of this study was to investigate FeNO variations due to environmental or occupational exposures in respiratory healthy subjects. Overall, 14 hairdressers and 15 healthcare workers in Oslo were followed for 5 workdays. We registered the levels of FeNO after commuting and arriving at the workspace and after ≥3 h of work, in addition to symptoms of cold, commuting method, and hair treatments that were performed. Both short- and intermediate-term effects after exposure were evaluated. Environmental assessment of daily average levels of air quality particulate matter 2.5 (PM_2.5_), particulate matter 10 (PM_10_), nitrogen dioxide (NO_2_), sulphur dioxide (SO_2_), and ozone (O_3_) indicated a covariation in ozone and FeNO in which a 35–50% decrease in ozone was followed by a near 20% decrease in FeNO with a 24-h latency. Pedestrians had significantly increased FeNO readings. Symptoms of cold were associated with a significant increase in FeNO readings. We did not find any FeNO increase of statistical significance after occupational chemical exposure to hair treatments. The findings may be of clinical, environmental and occupational importance.

## 1. Introduction

Nitric oxide (NO) was first detected as an intracellular messenger in various cells, such as in platelets, the nervous system, and vasculature, where it is known for its relaxing activity related to the endothelium and vasodilation and as an effector molecule in immunological reactions [1]. It was later measured in exhaled breath as fractional exhaled NO (FeNO) and was shown to be increased in patients with asthma [2,3]. Additionally, NO has been implicated in several inflammatory diseases, obesity, diabetes, and heart disease [4]. NO is regulated by nitric oxide synthase, and three major isoforms have been identified, of which inducible nitric oxide synthase (iNOS or NOS2) is associated with immunoregulation by cytokines and other stimuli in both the innate and the adaptive immune system [5,6]. Activated macrophages and other innate immune cells generate NO as a pro-inflammatory response to various pathogens [7]. More recent research has shown that iNOS is regulated on the epigenetic level by DNA methylation after environmental and occupational exposures [8,9,10]. In asthma, iNOS has predominantly been associated with the regulation of T-cell function and differentiation [11]. High FeNO values are associated with allergic/eosinophilic inflammation, also known as Type 2 inflammation [12]. FeNO can be used as an indicator of inhaled corticosteroid response and has been present in clinical use for several years as an evaluation tool for asthma control [13]. Diurnal variations of FeNO have been recorded in the airways of healthy subjects from roughly 5 to 20 ppb (parts per billion), in controlled asthmatics from 20 to 40 ppb, and in uncontrolled asthmatics from 20 to 70 ppb [14].

Previously, FeNO was found to be significantly elevated in a cohort of welders at levels that were normally associated with Type 2 inflammation (median 43.5 ppb) [15]. Welders are exposed to respiratory irritants (particulate matter, gases and smoke). The occupational hazards in a hairdressing salon are complex and include many respiratory irritants and allergens [16]. To our knowledge, there are no previous studies investigating FeNO variations after exposure in hair salons.

Increased air levels of particulate matter 2.5 (PM_2.5_), particulate matter 10 (PM_10_), nitrogen dioxide (NO_2_), sulphur dioxide (SO_2_), and ozone (O_3_) exacerbate asthma in children and adults and are associated with the onset of childhood asthma [17]. Normal and increased levels of FeNO are difficult to interpret in both the diagnosis and treatment of asthma [18]. FeNO variations require further explanation with respect to their role in identifying and treating respiratory diseases and environmental and occupational exposure.

The aim of the study was to investigate short- and intermediate-term FeNO variations after environmental and occupational exposures in hairdressers and healthcare workers (HCWs).

## 2. Materials and Methods

### 2.1. Study Design

Non-smoking subjects that were aged 18 years or older and scheduled for 5 working days each week were included in this study. The exclusion criteria were active smoking and physician-diagnosed respiratory diseases. The study was set up as an observational study of hairdressers and HCWs. A total of 15 hairdressers working at six hair salons in downtown Oslo covering an area of about one square kilometer (~0.4 square miles) were recruited via invitation. However, one hairdresser was considered to be an outlier (FeNO > 60 ppb) due to a probable respiratory disease. A total of 15 HCWs working at outpatient clinics or as technical assistants were recruited from Oslo University Hospital at two different locations 3–6 km (1.9–3.7 miles) from the downtown area. The HCWs were both an occupational control, with respect to the hairdressers, and an environmentally exposed group, as they experienced similar exposures as the hairdressers by living in or close to Oslo. The study was performed over two consecutive work weeks due to logistics and to minimize sensor variations by using the same FeNO testing instrument sensor. A questionnaire was included in the case report form, in which daily questions were asked and noted by the investigators at all sampling times. Daily questions included symptoms, mode of commuting and traveling time, exposure to smoke or vaping fumes, and the type and number of hair treatments that were performed. Most hairdressers started their workday 09:00–11:00 a.m., whereas HCWs started their workday 08:00–09:00 a.m. Due to sampling logistics, a few hairdressers started their day before the first sampling; however, the samples were taken within the first 60 min of work, and the occupational exposures in these instances were regarded as small. The daily sampling of the hairdressers followed their usual work week from Monday to Saturday with one day off. Only four hairdressers worked Saturday. All of the HCWs worked Monday to Friday.

A network of air pollution detectors is easily accessible in Norway that provides information on per-minute, average hour, and average daily levels of PM_2.5_, PM_10_, NO_2_, SO_2_, and ozone (www.nilu.no (accessed on 29 December 2022)).

### 2.2. FeNO Measurement and NIOX VERO© Repeatability

A portable NIOX VERO^®^ device was used to measure the level of FeNO in ppb. It had a disposable mouthpiece with a filter. A fresh pre-calibrated sensor that included 300 measurements was used for the whole study and during the repeatability tests. FeNO was measured after exhalation, followed by inhalation through the mouthpiece and a NO-scrubber for NO-free air supply, and lastly, by exhalation trough the mouthpiece with a respiratory rate of 50 mL/s (±5 mL/s) for 10 s. There was one unexpectedly high within-day change (~20 to 40 ppb) of measurement that was repeated with a fresh mouthpiece, with only one ppb change. To test for instrument repeatability, four subjects in the HCW group were selected. They performed five additional continuous FeNO measurements with fresh mouthpieces (the measurements are shown in Appendix A). The coefficient of variation (CV) was calculated as within-subject SD/within-subject mean, as described previously [19]. Variation was estimated to be 8.5% ± 3.5%. The manufacturer of NIOX VERO© provides the following precision data: <3 ppb of measured value for values <30 ppb, <10% of measured value for values ≥30 ppb.

### 2.3. Statistical Analysis

FeNO, as the primary end-point, was tested as a continuous variable, in addition to sampling and commuting time. Normality distribution tests (Kolmogorov–Smirnov and Shapiro–Wilk) showed both normality and non-normality for FeNO and other key variables. We applied both parametric tests with Student’s *t*-test unpaired and paired with Levene’s test for equality of variances and a non-parametric test with Mann–Whitney and histogram distribution. The statistical analyses were performed by using SPSS v28 and Graphpad Prism 9.2.0.

### 2.4. Ethics

All of the participants provided informed consent, and we can confirm that all of the research was performed in accordance with relevant guidelines/regulations. The study was approved by the regional ethics committee (case no. 480861), the Hospital Data protection officer (case no. 22–16786), and registered at www.clinicaltrials.gov (accessed on 29 December 2022) (Identifier: NCT05507944).

## 3. Results

### 3.1. Demographics

The HCWs were significantly older than the hairdressers (45.9 vs. 33.4 years *p* = 0.001) (Table 1). Otherwise, there were no significant demographic differences between the two groups.

### 3.2. Diurnal Variation of FeNO

Figure 1a,b shows the diurnal variations in FeNO during week 38 (19–24 September 2022) and week 39 (26–30 September 2022) among hairdressers and HCWs. No short- or intermediate-term increase in FeNO was detected during the weeks.

### 3.3. Air Quality Levels

Figure 2a shows the corresponding average daily air quality levels. The data values are available in Appendix A. A decrease in the ozone levels during both weeks corresponds with a decrease in FeNO S (measured after commuting and arriving at the workspace) but not in FeNO E (measured after ≥3 h at work), with a 24-h latency. In Figure 2b, the decrease in ozone and FeNO S in ppb is emphasized. When ozone decreased by 48% or 9.38 ppb (19.73 to 10.35 ppb) in week 38 and 34% or 9.33 ppb (27.75 to 18.42 ppb) in week 39, FeNO S decreased by 19% or 3.25 ppb (17.25 to 14.00 ppb) and 20% or 3.75 ppb (18.46 to 14.71 ppb), respectively. The FeNO values are available in Appendix A. When correcting for symptoms of cold, as shown in Appendix A, FeNO S decreased by 26% or 4.89 ppb (19.09 to 14.20 ppb) and 12% or 1.98 ppb (16.25 to 14.27 ppb), respectively.

### 3.4. FeNO Measurements, Sampling and Commuting Time

Among hairdressers and HCWs, there was no significant daily increase in FeNO (Table 2). The distribution of the data and normality tests are available in Appendix A.

### 3.5. Symptoms of Respiratory Infections

There were significantly increased FeNO S and FeNO E measurements among the participants that reported cold symptoms (*p* < 0.001) (Figure 3). No participants reported fever or shortness of breath.

### 3.6. FeNO, Commuting and Hair Treatments

Figure 4a shows the FeNO measurements related to commuting by car, public transport, bicycle, and as pedestrians. The data values are available in Appendix A. There was a significant increase in FeNO S and FeNO E among those who reported commuting as pedestrians (*p* = 0.026 and *p* = 0.040). Figure 4b shows the FeNO measurements in relation to the hair treatments that were performed by hairdressers, including bleaching, dyeing, the use of hair spray and other treatments (mostly nails). The data values are available in Appendix A. Among those who reported that they had performed bleaching and dyeing, there was a significant decrease in FeNO S (*p* = 0.007 and *p* = 0.009) and FeNO E (*p* = 0.014 and *p* = 0.007).

## 4. Discussion

We did not detect any short- or intermediate-term increases in FeNO corresponding to occupational exposures among the hairdressers. All of the hair salons had good ventilation systems. Hairdressers performing bleaching and dyeing had the lowest FeNO levels. The cause is unclear, although the non-exposed groups had higher than normal levels of FeNO. As expected, symptoms of cold significantly increased FeNO. FeNO was slightly increased among those who commuted as pedestrians for 5 min or more. They may also have been more exposed to air pollution than other commuters. However, FeNO did not increase among those who bicycled, although this was a small group. Interestingly there were large decreases in FeNO after commuting on different weekdays for both hairdressers (week 38, Wednesday to Thursday) and HCWs (week 39, Thursday to Friday). When compared with air pollution levels, large decreases were also present for ozone during both weeks, although one day before the decrease in FeNO. The decrease in ozone was 35–50%, whereas the decrease in FeNO was close to 20%, which is above our repeatability analysis and the NIOX VERO© manufacturer precision data.

Several studies have investigated FeNO and occupational exposures over the last 20 years with varying results, showing elevated values in studies focusing on spray painters, underground tunnel workers and welders, among others [15,20,21]. Daily increases in FeNO, up to 40%, in shoe and leather makers were found in one study [22]. Our findings may explain unexpected variations in FeNO measurements. Welding, spray-painting, and shoemaking, which produce ozone or volatile organic compounds (VOCs), an ozone precursor, increase FeNO levels among the exposed workers. These studies share large differences in baseline values of FeNO, ranging from 6 ppb to 25 ppb. A recent systematic review of occupational asthma and FeNO noted that different threshold levels of FeNO made drawing conclusions difficult [23]. Occupational studies have focused on particle matter exposures and chemicals without accounting for environmental effects. It is common in occupational medicine and hygiene to expect exposure in the workplace to be several times, if not magnitudes higher, than ambient environmental levels. For example, in the study related to welders [15], the median of PM_2.5_ was 604 µg/m^3^, and the highest air pollution level in Oslo during the two weeks in our study of PM_2.5_ was 10 µg/m^3^. The other markers of air pollution during our study were relatively low, and they did not correlate with the FeNO measurements, although NO_2_ showed some inverse correlation with ozone.

Ozone is different in this regard, as ambient environmental and occupational levels can be in the same range. Ground-level ozone is produced through chemical reactions between solar radiation, nitrogen oxide pollution (NO_x_) and VOCs [24]. Air pollution levels of ozone can reach more than 100 ppb in polluted areas, whereas welding in occupational settings can reach close to 200 ppb [25]. The Occupational Safety and Health Administration (OSHA) standard for ozone is 100 ppb averaged over eight hours, whereas The World Health Organization (WHO) sets an eight-hours environmental limit of 50 ppb [26,27]. Indoor levels of ozone in offices are about 10% of outdoor levels and may, by emission from printers and photocopiers, increase to 30–40% of outdoor levels [28].

Ozone and FeNO covariation were only present in the FeNO sampling after commuting and not after ≥3 h of work, which may indicate, although hypothetically, that other forms of exposure, such as chemical, biological, or physical, possibly have short-term effects depending on agent and dose.

A study on twins concluded that environmental contributions accounted for 40% of FeNO variations; the remaining was related to genetics [29]. A community-based population study comparing FeNO measurements and air pollution exposure showed a positive association of ozone in non-asthmatics with a five-day average air pollution of ozone [30]. In a longitudinal study of an elderly population, 12 weeks of weekly FeNO measurements showed a positive correlation between FeNO and five-day average ozone air pollution [31]. However, the effect of exposure to 300 ppb ozone on healthy volunteers for 75 min did not show any increased FeNO at 6 and 24 h post-exposure [32]. Studies on daily variations of FeNO in healthy subjects have found small within-day and between-day changes [19,33]. We have not found any studies in environmental or occupational settings that have performed diurnal FeNO measurements concerning variations.

Epidemiological studies of airway disease and exposure to ozone have not been consistent, possibly because ozone is a secondary air pollutant that is confounded by NO_x_. However, there is evidence that supports the correlation between exposure to ozone and childhood-onset asthma [34]. A recent, large case–control study showed that exposure to ozone was the only air pollution that was associated with asthma exacerbation requiring hospitalization [35].

A biological mechanism of the pathological effects of ozone is suggested in a mouse model of rhinitis [36]. Lymphoid cell-sufficient mice that were exposed to 500 ppb ozone for 4 h daily up to 9 days developed nasal Type 2 immunity and eosinophilic rhinitis with mucous cell metaplasia, whereas lymphoid cell-deficient mice did not. A marked influx of neutrophils was detected 2 h post-exposure but less after 24 h, and eosinophils dominated after 4 and 9 days. Several animal and human studies have supported an airway remodeling effect by long-time ambient ozone exposure [37,38].

The strengths of our study are that it used the diurnal measurements of workers that were exposed to airway irritants during a whole work week, both before and after occupational exposure, in addition to the consideration of environmental exposures during two work weeks. The limitations are a relatively low number of workers in each occupational group and that it is a relatively short longitudinal study concerning more subtle environmental effects. In addition, except for hair treatments, commuting method, and time, we obtained no data related to indoor and outdoor exposures during leisure time.

## 5. Conclusions

We did not find any short- or intermediate-term increases in FeNO in hairdressers related to work exposure. There was a decrease among workers that were exposed to bleach and dyeing, although the significance of this is unclear. There were variations in FeNO among both hairdressers and HCWs, which may be attributed to environmental ambient ozone levels. These are intriguing findings of clinical, environmental, and occupational importance, which need to be followed up by larger diurnal cohort studies in both respiratory healthy and non-healthy individuals.

## Figures and Tables

**Figure 1 ijerph-20-04271-f001:**
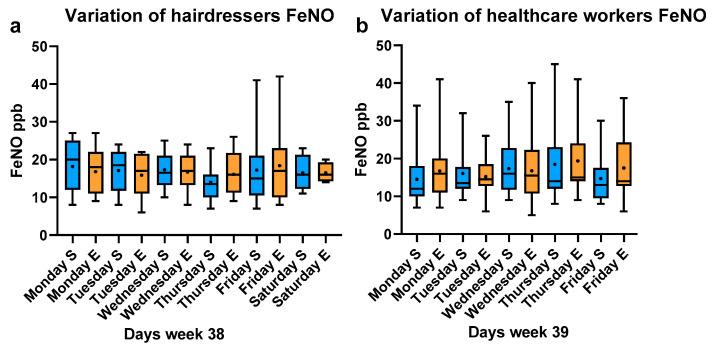
Diurnal variation in fractional exhaled nitric oxide (FeNO) in hairdressers and healthcare workers (HCWs). The dot in the boxplot shows the mean. The line shows the median. The box shows 25th and 75th percentile. The bar shows min. and max. After commuting and arriving at workplace (S). After ≥3 h work (E). Diurnal FeNO measurements on hairdressers during week 38, Monday to Saturday (**a**), and on HCWs during week 39 Monday to Friday (**b**).

**Figure 2 ijerph-20-04271-f002:**
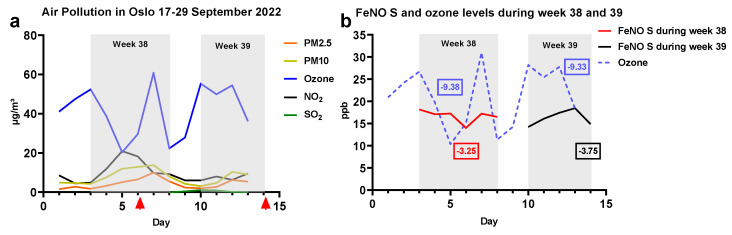
Air pollution levels and daily variation in fractional exhaled nitric oxide (FeNO). (**a**) Daily average air pollution levels in µg/m^3^ of PM_2.5_, PM_10_, Ozone, NO_2_, and SO_2_ recorded at Sofienbergparken monitoring station in Oslo during 17–29 September 2022. The station did not record 30 September. Monday to Saturday during week 38 and Monday to Friday during week 39 are marked with a grey background. Red arrows on the x-axis are days with the lowest FeNO measurements. (**b**) Comparison of ozone and FeNO S (after commuting and arriving at workplace) variation in ppb with emphasis on lowest levels and measurements during weeks 38 and 39.

**Figure 3 ijerph-20-04271-f003:**
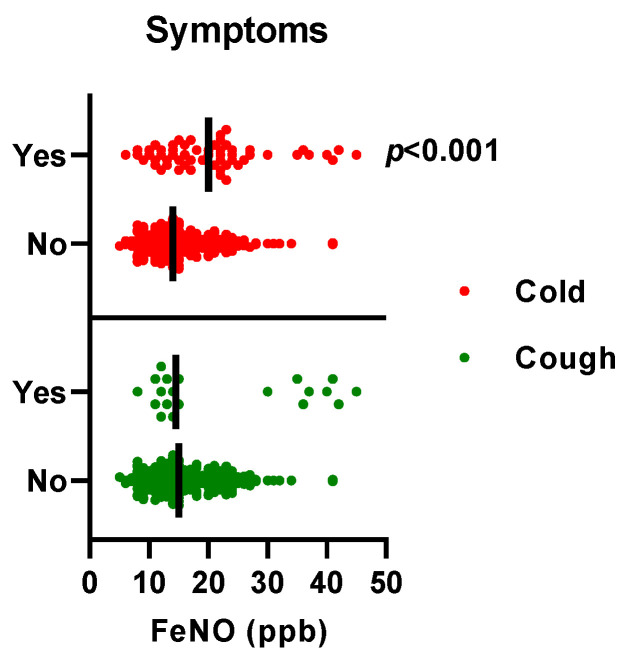
Fractional exhaled nitric oxide (FeNO) and symptoms of respiratory infection. Self-reported symptoms of respiratory infection, cough, and cold and FeNO S and FeNO E measurements during weeks 38 and 39 in both hairdressers and healthcare workers. The line shows the median. *p*-values are calculated with a Mann–Whitney test.

**Figure 4 ijerph-20-04271-f004:**
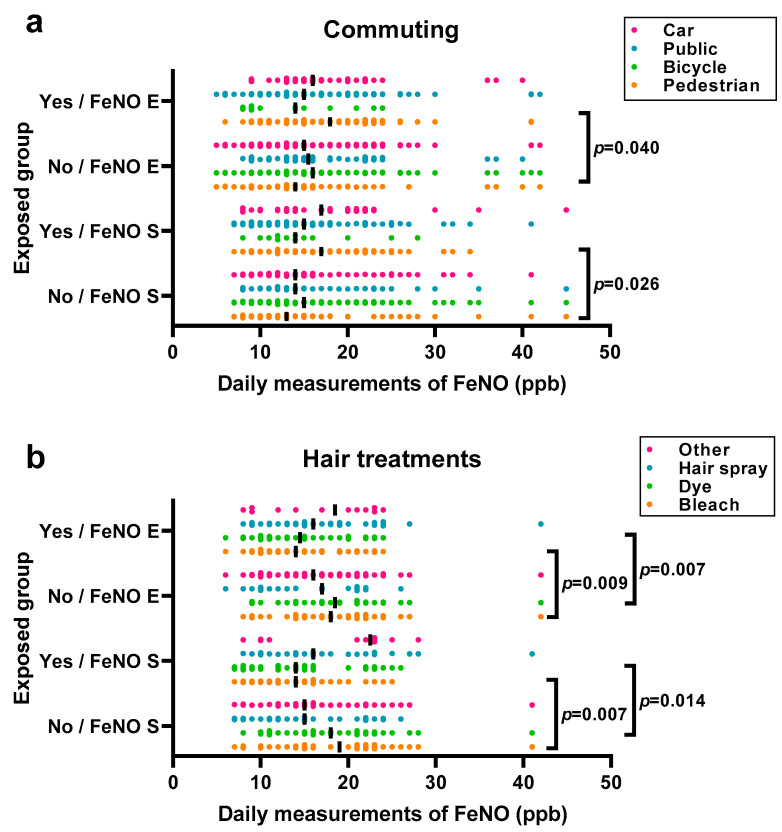
Fractional exhaled nitric oxide (FeNO), commuting and hair treatments. After commuting and arriving at workplace (FeNO S). After ≥3 h work (FeNO E). The line shows the median. (**a**) Comparison of hairdressers and healthcare workers’ daily FeNO measurements after commuting by car, public transport, bicycle, and as pedestrians. *p*-values are calculated with a Mann–Whitney test. (**b**) Comparison of hairdressers’ daily FeNO measurements by hair treatments with bleaching, dyeing, permanent, or other treatments. *p*-values are calculated with a Student’s *t*-test.

**Table 1 ijerph-20-04271-t001:** Demographics of hairdressers and healthcare workers.

	Hairdressers *N* = 14			Healthcare Workers *N* = 15			
Female/Male (*N*)	12/2			12/3			
Atopy (*N*)	3			2			
	Mean	SD	Range	Mean	SD	Range	*p*-value *
Age (Y)	33.4	7.3	21–50	45.9	12.5	22–66	**0.001**
Weight (Kg)	70.3	14.3	48–95	70.8	13.3	50–93	0.460
Height (Cm)	166.5	5.7	158–174	170.2	8.7	160–190	0.093
Body Mass Index (Kg/M^2^)	25.2	4.5	18.7–33.7	24.0	3.3	19.5–30.8	0.224
Hairdresser Years	12.7	5.9	2–26				
Saloon Employees (*N*)	6.3	1.1	3–7				
Saloon Area (M^2^)	103.3	30.9	60–140				

* *p*-values are calculated with Student’s *t*-test. Bold shows significance.

**Table 2 ijerph-20-04271-t002:** Hairdresser and healthcare workers fractional exhaled nitric oxide (FeNO) sampling and commuting time.

	Feno S	Feno E	*p*-Value	Sampling *	Commuting *
Hairdressers					
*N*	62	61		61	62
Mean	16.71	16.82	0.456	211.9	23.5
Median	16.00	17.00		194.0	25.0
Sd	6.5	6.3		39.7	10.7
Range	7–41	6–42		180–337	1–55
Healthcare workers					
*N*	70	70		70	70
Mean	16.17	17.09	0.315	220.0	41.0
Median	13.50	15.00		194.5	30.0
Sd	7.5	8.2		41.7	28.0
Range	7–45	5–41		180–317	4–120

* Time in minutes. After commuting and arriving at workplace (FeNO S). After ≥3 h work (FeNO E). *p*-values were calculated with a Mann–Whitney test.

## Data Availability

Anonymous datasets will be made available upon reasonable request by B. Hammarström.

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
