# Peer review of "Ambient Environmental Ozone and Variation of Fractional Exhaled Nitric Oxide (FeNO) in Hairdressers and Healthcare Workers"

_ijerph, 2023, doi:10.3390/ijerph20054271_

Round 1
Reviewer 1 Report
The manuscript entitled "Ambient environmental ozone and variation of fractional ex- 2 haled nitric oxide (FeNO) in hairdressers and healthcare work- 3 ers" by Tonje Trulssen Hildre et al describes The study investigated variations in FeNO levels in respiratory healthy subjects, hairdressers, and healthcare workers due to environmental or occupational exposures over 5 workdays. Results showed a decrease in FeNO after a decrease in ozone levels and an increase in FeNO with symptoms of cold. No significant increase in FeNO was found after exposure to hair treatments. These findings have potential implications for clinical, environmental, and occupational health.
The language in the manuscript is clear and concise.
The introduction can be improved by adding on the recent study on iNOS mechanism.
This is an interesting study that looks at the relationship between FeNO (fractional exhaled nitric oxide) levels and occupational and environmental exposures. The study highlights the findings of previous studies investigating FeNO levels in various occupational settings.
The study provides its strengths, which include diurnal measurements of workers exposed to airway irritants during a workweek, both before and after occupational exposures, as well as consideration of environmental exposures during two workweeks.
However, the study acknowledges its limitations, including a relatively low number of workers in each occupational group, a relatively short longitudinal study concerning more subtle environmental effects, and the lack of data on indoor and outdoor exposures during leisure time, except for hair treatments, commuting method, and time.
Overall, the study provides valuable insights into the relationship between FeNO levels and occupational and environmental exposures. However, further research is needed to fully understand this relationship and to determine the best approach for measuring and interpreting FeNO levels in different settings.
Author Response
Reviewer 1 wrote:
The introduction can be improved by adding on the recent study on iNOS mechanism.
Our reply:
We thank the reviewer for the comments.
The reviewer noted that the introduction must be improved and this was specified by adding on the recent study on iNOS mechanism. The exact mechanism was not specified by the reviewer. We have not described the molecular mechanisms behind regulation of iNOS transcription in the introduction, i.e. Mitogen Activated Protein Kinases (MAPK) pathways and NFkB, which are molecules and pathways which the author has done molecular research on previously. The reason not to describe the molecular mechanism was to keep the introduction concise at a clinical level. For example, the MAPK-pathway interacts with the major intracellular signaling pathways and it is not easily described. However, we assess that the reviewer is referring to more recent research that show an epigenetic mechanism that regulates iNOS after environmental and occupational exposures. We agree that this is an important finding, which should be mentioned in the introduction because of its relation with the aims of the study. We apologies to the reviewer if the reviewer was referring to another iNOS mechanism and we will certainly consider it if needed.
Reviewer 2 Report
Dear authors, please see the attached file.

Reviewer 3 Report
This is an interesting manuscript on an important issue. I have a few comments.
1. There are a large number of abbreviations that hamper legibility. I suggest you spell out the abbreviations, other than PM2.5 etcetera.
2. Figure 2 is hard to follow. You refer ti a decrease of 26% and 12%. Where in the figure do they occur?
3. Page 5, line 158: it says '... and FeNO E among participants' should behaps be '... and FeNO E levels among ...'.
Author Response
Reviewer 3 wrote:
- There are a large number of abbreviations that hamper legibility. I suggest you spell out the abbreviations, other than PM2.5 etcetera.
We thank the reviewer for the comments.
Our reply:
We agree with the reviewer that there are a large number of abbreviations. We have tried to address this issue by spelling out ozone in addition to spell out the abbreviations first used in each table and figure as suggested by reviewer 2. We hope this has improved the manuscript.
- Figure 2 is hard to follow. You refer to a decrease of 26% and 12%. Where in the figure do they occur?
Our reply:
The reviewer pointed out that figure 2 was hard to follow and was missing information of the data. Figure 2 b has been reworked and the data is now consistent with the figure, text in both manuscript and the supplement file. The text has been improved and the correct table in the supplement file has been renamed.
- Page 5, line 158: it says '... and FeNO E among participants' should behaps be '... and FeNO E levels among ...'.
Our reply:
The reviewer pointed out the word “level” was missing after FeNO. This has been noted at several places in the manuscript and been corrected. Hopefully, this will improve the readability.